# Investigation of the Efficacy of a *Listeria monocytogenes* Biosensor Using Chicken Broth Samples

**DOI:** 10.3390/s24082617

**Published:** 2024-04-19

**Authors:** Or Zolti, Baviththira Suganthan, Sanket Naresh Nagdeve, Ryan Maynard, Jason Locklin, Ramaraja P. Ramasamy

**Affiliations:** 1Nano Electrochemistry Laboratory, College of Engineering, University of Georgia, Athens, GA 30602, USA; or.zolti@uga.edu (O.Z.); baviththira.suganthan@uga.edu (B.S.); sanket.nagdeve@uga.edu (S.N.N.); 2Department of Chemistry, University of Georgia, Athens, GA 30602, USA; rkmaynard@uga.edu (R.M.); jlocklin@uga.edu (J.L.)

**Keywords:** food safety, *Listeria monocytogenes*, bacteriophage, impedance, pathogen detection

## Abstract

Foodborne pathogens are microbes present in food that cause serious illness when the contaminated food is consumed. Among these pathogens, *Listeria monocytogenes* is one of the most serious bacterial pathogens, and causes severe illness. The techniques currently used for *L. monocytogenes* detection are based on common molecular biology tools that are not easy to implement for field use in food production and distribution facilities. This work focuses on the efficacy of an electrochemical biosensor in detecting *L. monocytogenes* in chicken broth. The sensor is based on a nanostructured electrode modified with a bacteriophage as a bioreceptor which selectively detects *L. monocytogenes* using electrochemical impedance spectroscopy. The biosensing platform was able to reach a limit of detection of 55 CFU/mL in 1× PBS buffer and 10 CFU/mL in 1% diluted chicken broth. The biosensor demonstrated 83–98% recovery rates in buffer and 87–96% in chicken broth.

## 1. Introduction

There are 17 different species of the *Listeria* genus. Among them, only two species are pathogenic: *Listeria ivanovii*, found almost exclusively in ruminants, and *Listeria monocytogenes* (*L. monocytogenes*), which can infect humans and cause illnesses [1,2]. *L. monocytogenes* is a facultative, anaerobic, Gram-positive, rod-shaped bacterium known since 1924. It is a psychrophile pathogen capable of multiplying at refrigeration temperature (4 °C) and surviving at temperatures as low as −17 °C, with an optimum growing temperature range of 30 to 37 °C [3,4]. *L. monocytogenes* infection leads to illnesses such as listeriosis, sepsis, myocarditis, meningitis, encephalitis, bacteremia, and intrauterine or cervical infections in pregnant women that could lead to miscarriages or stillbirth [5]. The most common path of *L. monocytogenes* infection is through the gastrointestinal tract, similar to other foodborne pathogens. It can be found in various food products like poultry, pork, beef, dairy products, bread, fish, ready-to-eat foods, and fresh produce [6]. Its ability to form biofilms facilitates infection from surfaces, transport vehicles, and stainless-steel appliances [2,7]. Liquid and semi-liquid products like broth, milk, or soft cheeses are suitable growing grounds for *Listeria* detection. Chicken broth is made by cooking chicken and raw vegetables in water and, therefore, can represent samples of chicken, vegetables, and liquid samples [7,8,9,10,11].

The detection of *L. monocytogenes* has been performed with various types of biosensors. As their name suggests, optical biosensors provide an optical signal through luminescence, fluorescence, or color. The optical approach is very sensitive and selective but requires expensive optical equipment and is sensitive to environmental interference [12]. Thermal biosensors measure the heat change due to bioreaction between the biorecognition molecule and specific analytes that correspond with the target pathogen. The method is fast and sensitive, but its selectivity is low due to non-target responses [13]. Electrochemical biosensors (ECBSs) measure the changes in electric parameters like the current, potential, or impedance of a system due to biological interaction between the target analyte and the biorecognition molecule on the working electrode. ECBSs are sensitive, selective, fast, low cost, and do not require trained personnel. They require a small sample volume and can be portable, which makes them optimal for home or field detection in buffer or food samples [6,12,14,15].

In our previous work, we developed different phage-based approaches for separating and detecting different pathogens [16,17,18,19]. A biosensor utilizing phage-immobilized quarternized carbon nanotubes (q-CNTs) for the detection of *L. monocytogenes* in 1× phosphate-buffered saline (PBS) with a limit of detection of 8.4 CFU/mL was also presented [3]. This work presents an adaptation of our biosensor to a portable platform constructed from commercially available screen-printed electrodes in order to detect *L. monocytogenes* in chicken broth samples. This newly adapted platform, as seen in Figure 1a, offers the ability to detect the pathogen without specialized lab equipment.

The most common detection methods for *L. monocytogenes* include genomic and antibody-based approaches. Electrochemical detection systems relying on antibodies for *L. monocytogenes* have demonstrated a limit of detection (LOD) of 35 CFU/mL in 1× PBS buffer and 22 CFU/mL in spiked lettuce samples. Despite their high sensitivity, antibody-based systems are constrained by limitations in stability and cost compared to bacteriophage-based alternatives [20,21]. Another detection strategy involves genomic methods, utilizing DNA or RNA for *L. monocytogenes* detection, and achieving remarkably low LODs on the order of 10^−14^ M. However, these methods have notable drawbacks, including the necessity of high-temperature sample preparation for denaturation and prolonged sample preparation times ranging from 8 to 24 h at minimum [22,23,24]. Additionally, nucleotide- or antibody-based methods are not capable of distinguishing living and dead bacterial cells and therefore are not particularly attractive for food safety testing. The phage-based method discussed in this work overcomes this drawback [25,26,27,28,29,30,31].

## 2. Materials and Methods

### 2.1. Materials Used

Carboxylic acid functionalized multiwalled carbon nanotubes (COOH-CNT) with a 30–50 nm outer diameter and a 10–20 µm length (from Cheap Tubes Inc., Cambridgeport, Vermont, USA); 1-pyrenebutanoic acid succinimidyl ester (PBSE), bovine serum albumin (BSA), Tween^®^ 20, dichloromethane and chlorodimethylsilane (all four from Sigma-Aldrich, St. Louis, MO, USA); dimethyl sulfoxide (DMSO) (Thermo-Scientific, Waltham, MA, USA); disodium phosphate (Na_2_HPO_4_) (Research Products International Corp, Mt Prospect, IL, USA); sodium chloride (NaCl) (EMD chemicals, Massachusetts, USA); magnesium sulfate heptahydrate (MgSO_4_·7H_2_O (J.T. Baker, Japan); thionyl chloride (SOCl_2_) and iodomethane (CH_3_I) (both from Alfa Aesar, Haverhill, Massachusetts, USA); potassium phosphate dibasic (KH_2_PO_4_) and potassium chloride (both from BDH, Solon, Ohio, USA); tris base, typtone, and ethanol (all three from Fisher Scientific, Hampton, New Hampshire); yeast extract and agar powder (both from Becton Dickinson and Company, Franklin Lakes, New Jersey, USA); and chicken broth (GreenWise ^®^, Lakeland, Florida, USA) were purchased from the respective commercial vendors and used as received. 

Phosphate-buffered saline 10× (100 mL) was prepared by mixing 0.2 g of KCl, 8 g of NaCl, 0.245 g of KH_2_PO_4_, and 1.4 g of Na_2_HPO_4_. PBS (1×) (pH 7.4) was prepared by diluting the PBS 10× buffer. A quantity of 0.01% tween 20 solution was prepared by mixing 10 mL of PBS (1×), 85 mL deionized water (DIW), and 20 µL of Tween^®^ 20. Luria Bertani (L.B.) (100 mL) (pH 7.0) was prepared by mixing 1 g of tryptone, 0.5 g of yeast extract, and 1 g of NaCl. S.M. buffer (pH 7.5) was prepared by mixing 100 mM NaCl, 8 mM MgSO_4_. 7H_2_O, 50 mM Tris base, and 0.01% gelatin. Standard brain–heart infusion (BHI) media was prepared by mixing 37 g of the BHI powder into 1 L of DIW using a magnetic stirrer until a homogenized solution was formed. Chicken broth, 1% dilution, was prepared by diluting 1 mL chicken broth into 99 mL 1× PBS and vortex-mixing the resulting solution. DIW with a resistivity of 18 MΩ.cm was used to prepare all the media and chemicals. All buffers and media were sterilized before use. 

Screen-printed electrodes (SPE) (Zensor, Taichung City, Taiwan) were purchased from C.H. Instruments, Inc., Austin, TX, USA and used as working-, counter-, and quasi-reference electrodes. All flow-based experiments were performed using a microfluidic flow cell from Metrohm Dropsens, Oviedo, Asturias. All electrochemical impedimetric measurements were performed using a CHI-920C model potentiostat (CH Instruments Inc., Austin, TX, USA).

### 2.2. Methods Used

#### 2.2.1. Microbiological Methods

*Listeria monocytogenes* Scott A, a pathogenic strain, was used as the target analyte, whereas *Salmonella enterica* subsp. *Enterica* serovar Typhimurium 291RH (ser. Typhimurium-291RH) and *Escherichia coli* O157:H7 (*E. coli* O157:H7) were used as the non-target pseudo-analytes for specificity studies. Listex P100 bacteriophage (P100 Phage) was purchased from Micreos Food Safety B.V, Wageningen, Netherlands. *L. monocytogenes* Scott A was grown by inoculating a single colony in 3 mL of BHI media and incubating at 37 °C for 24 h at 200 rpm. Both ser. Typhimurium-291RH and *E. coli* O157:H7 were grown by inoculating a single colony in 3 mL of BHI and Luria Broth (LB) media, respectively. Both cultures were incubated overnight for 18 h at 37 °C at 200 rpm. One mL of the mid-log phase bacterial culture was centrifuged at 5000 rpm for 8 min. For detection experiments in buffer, the supernatant was removed and washed twice with 1× PBS buffer to remove any media residue, and the pellet was resuspended in 1× PBS buffer. Then, the dilution series was prepared. For detection experiments in 1% diluted chicken broth, all supernatant was removed and 1× PBS buffer was used to wash and remove any media residue, and the pellet was resuspended in 1% diluted chicken broth. The dilution series was also prepared with 1% chicken broth as the media used. Enumeration of bacteria was performed by plate-count techniques and expressed in CFU/mL. A plaque assay was carried out with P100 phage and *L. monocytogenes* to measure the phage titer and was expressed in PFU/mL. A soft agar overlay technique was carried out to evaluate the specificity of the P100 phage towards the target (*L. monocytogenes*) and non-target bacteria (ser. Typhimurium-291RH and *E. coli* O157:H7), with the presence and absence of P100 phage.

#### 2.2.2. Electrode Preparation

Quaternized carbon nanotubes (q-CNT) were prepared according to the protocol presented by Zolti et al. [3]. Screen-printed electrodes (SPE) were rinsed with DIW and dried at room temperature for 2 h prior to modification with q-CNT. Once the electrode dried, 8 µL of the 1 mg/mL q-CNT solution was drop-cast on the SPE working electrode and then dried at room temperature. After that, PBSE as a molecular tethering agent was used as a crosslinker to attach the P100 phage to the q-CNT modified electrode. The modified SPE was rinsed with 1× PBS and placed in an ice container and 0.5 µL of 20 mM PBSE solution (in DMSO) was dropped onto it and allowed to self-assemble for 15 min. Excess PBSE was removed by rinsing twice with 1× PBS prior to phage attachment. One µL of the 10^9^ PFU/mL P100 phage solution was drop-cast on the working electrode and kept overnight at 4 °C. The P100 phage contains negatively charged capsids and positively charged tail fibers. The strong positive charge on the q-CNT created an oriented phage layer chemically anchored to the surface, as presented by Zolti et al. [3]. After immobilization, the electrode was rinsed with SM buffer and washed with 1× PBS buffer twice. Following the wash, 0.5 µL of 0.1% BSA solution was deposited on the electrode for 30 min to block areas that might not have been completely modified. Finally, the electrode was incubated in 1× PBS or with 1% diluted chicken broth for 15 min before use in electrochemical experiments. 

The bacterial solution (100 µL) was drop-cast onto the SPE and incubated for 8 min before the measurement. The impedimetric characterization was carried out using a CHI-920C scanning electrochemical microscope. The electrochemical system was a 3-electrode SPE, as shown in Figure 1a. Electrochemical impedance spectroscopy (EIS) measurements were performed in 5 mM [Fe(CN)_6_]^4−^/[Fe(CN)_6_]^3−^ as redox couple, with a frequency range of 1 Hz to 100 kHz and an AC amplitude of 5 mV. All measurements were performed at room temperature under standard conditions. The modified SPEs were tested in 1× PBS buffer and 1% diluted chicken broth matrices. The negative control contained no bacteria, and the test samples contained different concentrations of *L. monocytogenes*. The SPE was rinsed with 1× PBS after incubation with the tested solution. A 100 µL quantity of 5 mM [Fe(CN)_6_]^4−^/[Fe(CN)_6_]^3−^ solution was dropped on the SPE to cover the working, counter, and reference electrodes prior to impedimetric measurements. The negative control measurement was used as the baseline R_CT_ for each set of measurements presented in this section. Detection experiments under constant flow were performed using a syringe pump connected to a microfluidic flow chamber, as shown in Figure 1b. The chicken broth was diluted a hundred-fold to 1% in 1× PBS to produce a more homogenized sample. Due to the dilution, the heterogenous nature of the chicken broth is negated, while the relevant target bacteria concentration is not reduced below the limit of detection. The dilution was performed according to practices common in the field [20,21].

## 3. Results and Discussion

### 3.1. Detection of L. monocytogenes in Buffer and Broth

Initial impedimetric measurements were performed with the *L. monocytogenes* suspended in 1× PBS buffer at a concentration range of 10^2^ CFU/mL to 10^6^ CFU/mL. Measurements in triplicate were performed to determine the errors. Figure 2a presents a Nyquist plot with data collected from the 1× PBS buffer experiments. The calibration data with a baseline boxplot within the inset are shown in Figure 2b, along with the linear confidence limits, with a confidence level of 95%, showing that all points fall within the linear regime. It is visible that at higher concentrations, a larger error is calculated in the buffer.

The reason for this error determination is that one of the biosensors has reached saturation at lower analyte concentrations than the other two. In addition, the rate of signal change from the concentration of 10^4^ CFU/mL has slowed at different rates. Following the buffer experiments, detection experiments were performed in which the negative control and the bacterial solutions were suspended in chicken broth. The diluted broth was used to reduce the effects of inconsistencies in broth composition. The results of these experiments are shown in Figure 2c,d. The data suggest that exposure to the broth causes a significant reduction in the overall values of the R_CT_, even after baseline adjustment, with respect to the corresponding measurements in the buffer. Additionally, the chicken broth measurements showed lower calculated error for all measurements. The lowest concentration measured was 10^2^ CFU/mL. In addition, two methods were used to calculate the limit of detection (LOD). The first involved use of a linear regression method, and resulted in 55 CFU/mL in buffer and 10 CFU/mL in broth. The second method was 6σ. Here, the LOD is defined as anything higher than three standard deviations from the baseline, with a confidence level of α = 0.01. The values of three standard deviations from the baseline were 38.8 Ω in broth and 126.2 Ω in buffer. When using these numbers to calculate the limit of detection from the linear equation on the calibration curve, the value corresponds to 10 CFU/mL in broth and 300 CFU/mL in buffer [22]. A possible explanation for the improvement with broth samples is that the different salts and components reduce the charge transfer resistance of the system, in turn lowering readings and making them easier to detect above the noise level. Also, since the broth was diluted to 1%, the LOD in undiluted broth samples would be 10^3^ CFU/mL for both methods, which meets the requirements of most Western countries and is on par with other biosensors [6]. In addition, the recovery rate is a parameter that compares the concentration calculated from the calibration curves to the actual concentration placed on the biosensor, as shown in Table 1, and this further proves the predictability and accuracy of the biosensor [23].

The sensor’s recovery rate in broth has increased with the concentration, and in some concentrations, it has been better than when in a simple matrix, here, 1× PBS buffer. This suggests that the biosensor can detect *L. monocytogenes* and produce a reliable measurement of its concentration in the broth sample, with an accuracy of 88% to 96%. The recovery rate is calculated using Equations (1) and (2).
(1)$$C_{i}^{calculated}=\frac{\Delta R_{CT,i}-intercept}{slope}$$
(2)$$Recovery\;Rate=\frac{C_{calculated}}{C_{actual}}$$

Following these tests, the biosensor’s stability over time was tested. SPEs were prepared simultaneously and submerged in 1× PBS at 4 °C until tested after 1 h, 1 day, 1 week, and 2 weeks. For each mentioned time, triplicate impedimetric measurements were obtained with 10^2^ CFU/mL *L. monocytogenes* in broth. The variation in impedance signal from the initial value was calculated as the percentage change from the results obtained after 1 h, as shown in Figure 3. Since all of the electrodes had been prepared simultaneously under the same conditions, the results after 1 h were used as the reference initial value (100%), and all other results were compared with this reference value. By doing so, it is possible to compare the changes in the signal over time. The stability measurements showed that the response was reduced by 10% after a day but maintained stability. After a week, an overall 30% reduction was observed, and the error became larger, and after 2 weeks, the signal was only 40% of its original value. It can be reasonably concluded that the biosensor exhibits stable performance for over a week after phage immobilization when used to detect *L. monocytogenes* concentration around 10^2^ CFU/mL.

### 3.2. Role of Phage as Biorecognition Molecule

After testing the biosensor using varying concentrations of *L. monocytogenes* in broth, the effectiveness of the P100 bacteriophage as a biorecognition molecule was tested to ensure that the impedimetric signal for *L. monocytogenes* detection can be attributed to the presence of the biorecognition molecule. Two sets of SPE were used, one unmodified and the other immobilized with P100 bacteriophage, for the same duration. Figure 4 shows the impedimetric response of the biosensor with and without the P100 bacteriophage. The response showed that when modified with the phage, there was a significantly higher impedance signal, which also increased with the analyte concentration, while the SPE without the phage showed an almost constant response with little effect shown from the varying analyte concentrations. The results indicate that the impedance signal can be attributed to the selective binding of *L. monocytogenes* to P100 bacteriophage on the SPE.

### 3.3. Specificity of the Biosensor

The specificity of the phage-modified biosensor was tested by exposing the SPE with and without P100 bacteriophage to non-target pathogens *E. coli* O157:H7 and ser. Typhimurium-291RH. These bacteria were chosen since both are rod-shaped and commonly found in chicken products, alongside *L. monocytogenes* [32,33,34]. The first set of experiments tested 10^2^ CFU/mL of single pathogen in broth. The second set of experiments contained a sample containing *E. coli* O157:H7 and ser. Typhimurium-291RH in concentrations of 10^3^ CFU/mL each. In addition, *L. monocytogenes* was added at two different concentrations, 10^2^ and 10^3^ CFU/mL. The impedimetric response to a single pseudo-analyte was 10–20 ohms above the response from the negative broth control, while the response to *L. monocytogenes* was 75 ohms above it, as seen in Figure 5a; the response to pseudo-analytes is 13–26% of the response to *L. monocytogenes.* This suggests that the biosensor is very specific, and a positive response will only originate from bacteriophage-*L. monocytogenes* interaction. The biosensor without the phage biorecognition molecule showed an 8–10-ohm response from all test solutions and the control, demonstrating the effectiveness of the phage. In the interference study, shown in Figure 5b, the biosensor’s responses to broth samples with both pseudo-analytes, and with and without *L. monocytogenes*, are presented. A clear signal was measured when *L. monocytogenes* was present, even when at lower concentration than the pseudo-analytes. In these measurements, when the biosensor had no phage, the response was almost constant, without any dependency on the concentration of *L. monocytogenes*, which further emphasizes the specificity of the P100 bacteriophage, even in a multi-contaminate environment.

### 3.4. Detection under Flow

The final step was to conduct *L. monocytogenes* measurements in broth at different flow rates. The capability to detect *L. monocytogenes* in flow conditions is an important proof-of-concept on the path to portable electrochemical biosensors and the ability to integrate such a sensor into a production line. Initially, a 0.01% Tween^®^ 20 solution was flowed through the system. Then, an SPE was inserted into the flow cell, and the broth or bacterial sample was flowed on the biosensor’s surface for 8 min at a flow rate of 0.1 mL/min. After that, 5 mM [Fe(CN)_6_]^4−^/[Fe(CN)_6_]^3−^ solution was flowed on the SPE surface at a rate of either 0.5 mL/min, 1 mL/min, or 2 mL/min. By multiplexing 10 biosensors of this size in parallel, it will be possible to process a liter of broth per hour. While the redox-couple solution was flowing, impedimetric measurements were taken. A Nyquist plot with the results of the 0.5 mL/min flow rate is shown in Figure 6a, and the responses from all flow rates are presented in the bar chart in Figure 6b. The results show that without the phage, the response is an increase of 1–5 ohm, and the flow rate did not move these values outside of that range. With phage-modified SPE, the response shows a 10% decrease in signal every time the flow rate doubles. Even though there was a decreased signal, the signals all changed with the concentration of *L. monocytogenes*. These results demonstrate that the whole detection process can be accomplished under flow after the SPE is prepared and modified. These results show the capability of the system to work as a portable system. 

## 4. Conclusions

In this study, a highly sensitive electrochemical biosensor tailored for *L. monocytogenes* detection was developed and evaluated. The biosensor exhibited exceptional efficacy, demonstrating a remarkable LOD of 10 CFU/mL, surpassing the capabilities of most existing devices and displaying a sensitivity two orders of magnitude superior to PCR. Successful detection assays were conducted in chicken broth containing multiple pathogens under continuous flow conditions. The selectivity of the P100 bacteriophage was effectively demonstrated through exposure to a singular pathogen and interference studies. Moreover, the integration of SPEs and microfluidic channels showcased the portability of the system. With a demonstrated stability of up to one week, the biosensor proves suitable for various food-pathogen testing applications. Furthermore, validation using chicken broth as a representative food matrix underscored the robust performance of the biosensor platform. 

## Figures and Tables

**Figure 1 sensors-24-02617-f001:**
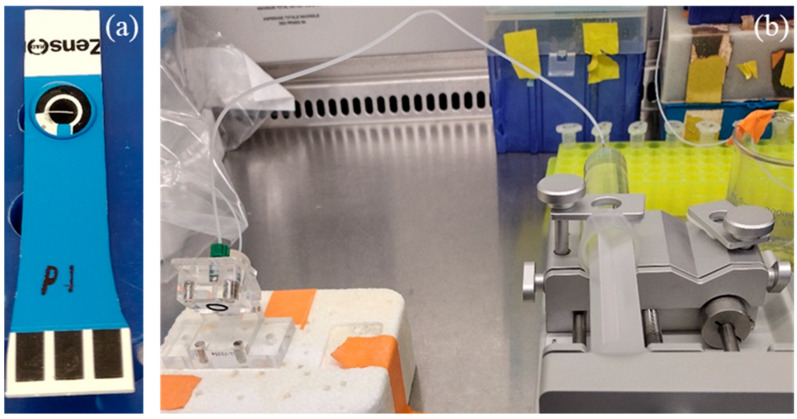
(**a**) SPE electrochemical biosensor; (**b**) Flow-based detection apparatus.

**Figure 2 sensors-24-02617-f002:**
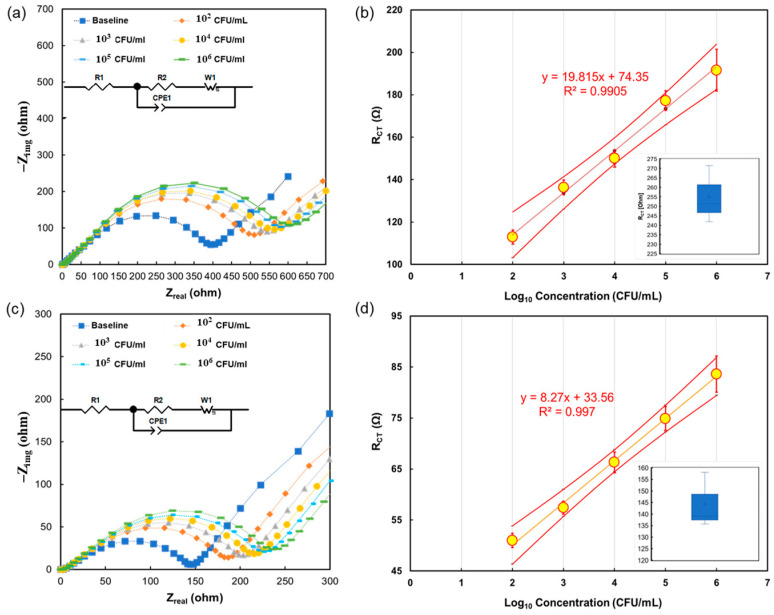
(**a**) Impedimetric response in 1× PBS to *L. monocytogenes*, with the equivalent circuit in the inset; (**b**) Reliable-range-of-calibration curve in 1× PBS buffer, along with linearity confidence interval limits, with a confidence level of 95%; the inset shows a box plot of the negative control (0 CFU/mL) measurements; (**c**) Impedimetric response in chicken broth to *L. monocytogenes*, with the equivalent circuit in the inset; and (**d**) Reliable-range-of-calibration curve in 1% chicken broth, along with linearity confidence interval limits, with confidence level of 95%; the inset shows a box plot of the negative control measurements.

**Figure 3 sensors-24-02617-f003:**
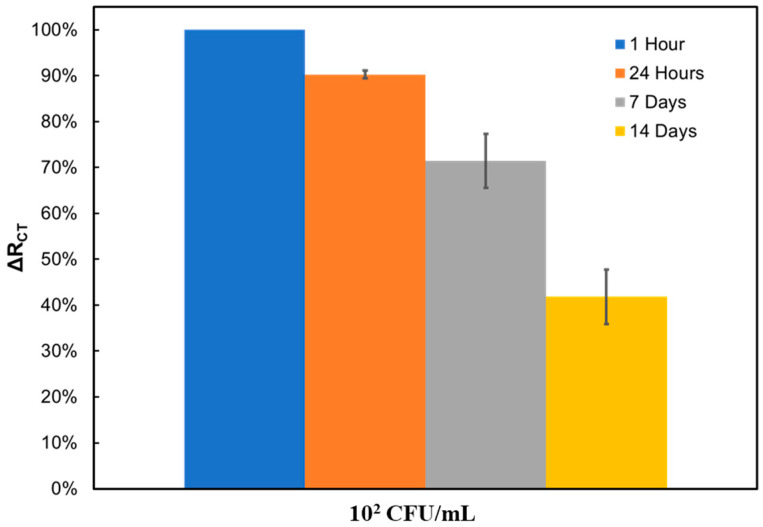
Biosensor stability over time, measured for broth.

**Figure 4 sensors-24-02617-f004:**
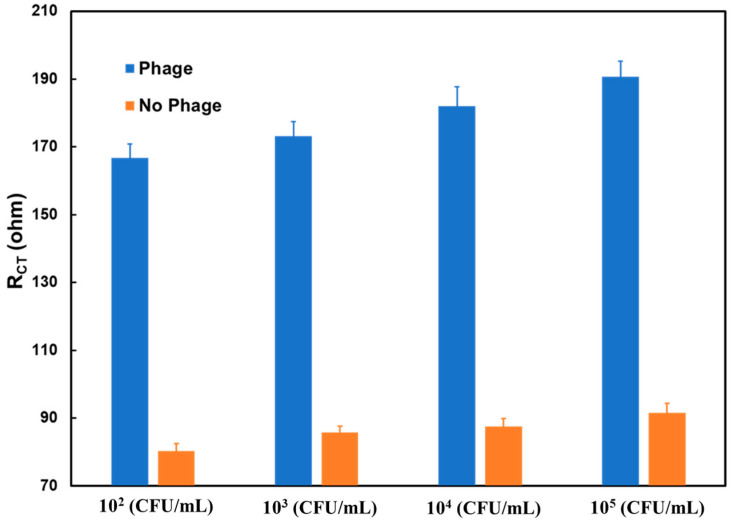
R_CT_ values from measurements of SPE modified with and without P100 bacteriophage, at different *L. monocytogenes* concentrations in a chicken broth sample.

**Figure 5 sensors-24-02617-f005:**
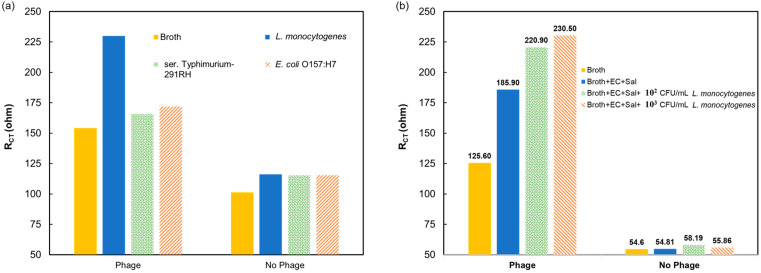
Specificity studies in 1% chicken broth: (**a**) biosensor responses to *L. monocytogenes* and non-target pathogens with and without P100 phage; and (**b**) interference study in which non-target pathogens were kept at 10^3^ CFU/mL and *L. monocytogenes* changes from 0 (negative control) to 10^3^ CFU/mL (EC = *E. coli* O157:H7 and Sal = ser. Typhimurium-291RH).

**Figure 6 sensors-24-02617-f006:**
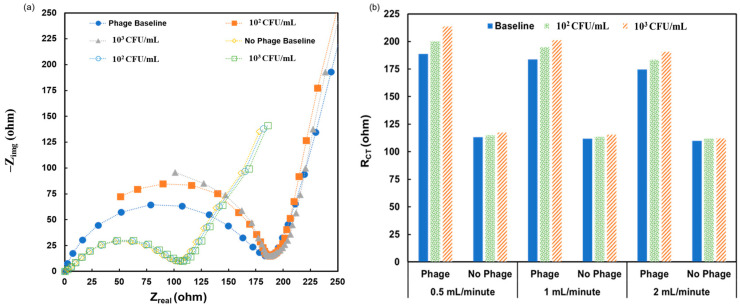
Biosensor’s response to *L. monocytogenes* at different concentrations under flow: (**a**) Nyquist plot of the response with a flow rate of 0.5 mL/minute; and (**b**) bar chart of the responses with different flow rates and different *L. monocytogenes* concentrations.

**Table 1 sensors-24-02617-t001:** Recovery rates in chicken broth and buffer.

Actual Concentration Log10 (CFU/mL)	Buffer		Chicken Broth	
	Calculated Concentration Log10 (CFU/mL)	Recovery Rate	Calculated Concentration Log10 (CFU/mL)	Recovery Rate
2	1.97	98%	1.75	88%
3	2.87	96%	2.77	92%
4	3.39	85%	3.78	95%
5	4.44	89%	4.80	96%
6	4.99	83%	5.75	96%

## Data Availability

Data are contained within the article.

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
