# Peer review of "Investigation of the Efficacy of a Listeria monocytogenes Biosensor Using Chicken Broth Samples"

_sensors, 2024, doi:10.3390/s24082617_

Round 1

Reviewer 1 Report

Comments and Suggestions for Authors

Sensors: 2946813

In this work, authors presented investigation of listeria monocytogenes using bacteriophage immobilized carbon nanotube-based nanostructured electrode. They demonstrated efficacy of Listeria monocytogenes in chicken broth. The biosensing platform was able to reach a limit of detection of 55 CFU/mL in 1X PBS buffer and 10 CFU/mL in 1 % diluted chicken 17 broth. The biosensor demonstrated 83 -98 % recovery rates in buffer and 87-96 % in chicken broth. This fabricated electrode is applicable in food industry and may benefit to public to identify the fresh food. However, I need some clarifications concerning several issues before publication. I therefore could not recommend present form of the manuscript for publication in sensor journal.

Comments:

1.      How long cultured Listeria monocytogenes can effectively use as analytes in normal temperature?  

2.       What is the novelty of developed work? Authors should add the performance of the developed electrode over other existing electrodes.

3.      Impedance responses of the electrode toward the detection of difference concentration of Listeria monocytogenes are not distinguishable, since some colors are almost similar. Authors should carefully use different colors and can be made difference with changing line width.

4.      I believe the fabricated electrode cannot detect lower concentration L. monocytogenes, since baseline detection curve is far from 10^2 cfu/ml. Authors should address in the revised manuscript about the harmful range of L.  monocytogenes for food poisoning.

5.      How did authors determine confidence level 95% of the measurements?  

6.      Abstract should be improved. and conclusion should be strong and supported to the results. 

Reviewer 2 Report

Comments and Suggestions for Authors

The authors have developed an electrochemical sensor for the detection of Listeria monocytogenes. The works follow a similar principle compared to previous works of the group. The main novelty declared by the authors is the use of a portable platform based on screen-printed electrodes. According to what it is explained in the work, the SPEs used are commercial and the potentiostat used is not included in the Materials section. Why is then the system portable?

The authors have picked several bacteria strains as analyte and non-target pseudo analyte. In the materials section, it is stated that bacteria samples were obtained from a 24-hour culture of bacteria starting with a single colony for inoculation. Then, 1 mL of each cultured was picked and the pellet was used for detection. But not all the bacteria would grow at the same rate. Have you checked the concentration for each of the strains used (checking optical density at 600 nm for example)? Although CFU counting is done, it is expressed in terms of the decimal unit, without considering the exact number of bacteria. Have concentrations and dilutions been performed making sure exactly 1 x 10^X CFU/mL were present in the sample? Considering the sensitivity of the technique, can you discuss how small bacteria concentration can influence the results obtained?

What concentration of P100 was used for electrode functionalization?

Can you explain more in detail how the buffer recovery assay has been designed? Is there any difference in the sample preparation between the buffer samples used for the calibration and the ones used for testing the recovery? It looks more like the data presented is just showing the variability of the method in between measurements than actually the recovery. The recovery is conventionally calculated by comparing a complex sample, in this case, the chicken broth, and the buffer. As in this case the calibration has been also done in chicken broth, I don’t see the point of this assay.

More discussion on how the sensor meets the requirements for the detection of L. monocytogenes in food samples is required, together with a better comparison with standard methods and alternative biosensors.

A conclusion section should be added.

The reference section needs to be corrected as some references are included several times.

Comments on the Quality of English Language

The quality of English is correct. Just minor revisions are required. 

Round 2

Reviewer 1 Report

Comments and Suggestions for Authors

I have carefully reviewed the revised version of manuscript and found that authors addressed all comments in the revised manuscript. I therefore, recommended for publication the present form of manuscript.  

Reviewer 2 Report

Comments and Suggestions for Authors

The manuscript is correct in the present form. Questions have been addressed and conclusion section has been added. 

Comments on the Quality of English Language

English is correct